# Prior-independent Dynamic Auctions for a Value-maximizing Buyer

**Yuan Deng**
Google Research
dengyuan@google.com

**Hanrui Zhang**
Duke University
hrzhang@cs.duke.edu

## Abstract

We study prior-independent dynamic auction design with production costs for a value-maximizing buyer, a paradigm that is becoming prevalent recently following the development of automatic bidding algorithms in advertising platforms. In contrast to a utility-maximizing buyer, who maximizes the difference between her total value and total payment, a value-maximizing buyer aims to maximize her total value subject to a return on investment (ROI) constraint. Our main result is a dynamic mechanism with regret $\tilde{O}(T^{2/3})$, where $T$ is the time horizon, against the first-best benchmark, i.e., the maximum amount of revenue the seller can extract assuming all values of the buyer are publicly known.

## 1 Introduction

Automatic bidding has become one of the main options for advertisers to buy advertisement opportunities in the online advertising market [Dolan, 2020]. The prevalence of automatic bidding is partly driven by the fact that it significantly simplifies the interaction between the advertisers and the advertising platform. Nowadays, millions of auctions are conducted in any given day while the competitive landscape could be evolving over time and becoming vastly different across auctions. Therefore, it is difficult for the advertisers to reason about its dynamics, let alone optimizing their bids for each auction separately. Instead of soliciting for fine-grained bids for different auctions they participate in, the automatic bidding product only asks the advertisers to provide high-level objectives and constraints. With provided high-level information, the automatic bidding product bids on behalf of the advertisers to optimize the objectives while respecting the constraints. While there are many different types of automatic bidding produces, a well-adopted product bids to maximize the number of acquisitions subject to a constraint on the advertiser-specified return on investment (ROI) constraint, i.e., a target ratio between the advertiser's return and her advertising spend [Facebook, 2021, Google, 2021]. To bid on behalf of the advertiser to maximize the number of acquisitions, the automatic bidding products can invoke sophisticated prediction algorithms to predict the acquisition rate. These prediction algorithms have the benefit of being trained with long periods of data to have a good generalization [Juan et al., 2016, McMahan et al., 2013, Zhou et al., 2018]. In this way, the advertiser can only focus on designing their high-level objectives and specifying their constraints, while leaving the complicated bidding part to the automatic bidding products.

The emergence of automatic bidding products opens up the opportunities for novel design of dynamic auction mechanisms, as classic mechanisms are heavily specialized for utility-maximizing bidders, the de facto paradigm in economic theory, which maximize the difference between their total value and total payment [Edelman et al., 2007, Myerson, 1981]. In contrast, automated bidders behave very differently in a way such that the payment is not directly involved in their objectives; and instead, the payment indirectly appears in the ROI constraints while the objectives aim to maximize the total value only. Therefore, classic auction design tailored for utility-maximizing bidders may not continue to perform well in an automatic bidding environment. Furthermore, recent results demonstrate that

35th Conference on Neural Information Processing Systems (NeurIPS 2021).

for single-slot auctions, neither welfare nor revenue achieved by an incentive-compatible auction (for utility-maximizing bidders, e.g., second-price auctions) at any reasonable equilibrium can be larger than $1/2$ of the optimum [Aggarwal et al., 2019, Deng et al., 2021]. Recently, Balseiro et al. [2021] initiated the study of characterizing single-stage revenue-optimal auctions for automated bidders under various information structures. Surprisingly, they show that in a Bayesian setting in which the advertiser' value is private and the target ratio in the ROI constraint is public, a payment-scaled second-price auction can achieve first-best revenue, i.e., the revenue when all information is public. For a comparison, recall that first-best revenue is in general not achievable against utility-maximizing bidders as demonstrated by the Myerson's auction [Myerson, 1981]. However, the assumption on perfect knowledge of the prior distribution on the advertiser's values limits the application of their payment-scaled second-price auction, as the seller usually needs to learn the distribution via repeated interactions with the advertiser.

## 1.1 Our results

In this paper, we aim to relax such an assumption by studying the problem of designing prior-independent dynamic mechanisms against a single automated bidder. In line with the literature [Amin et al., 2013], we consider a setting of repeated auctions with multiple stages against an impatient automated bidder whose valuations are drawn identically and independently across stages. Moreover, we assume the automated bidder bids in a way to satisfy her ex-ante ROI constraint per stage, consistent with the setup in [Balseiro et al., 2021]. In practice, the auction parameters are updated infrequently (e.g., once per day) and the automated bidders participate in multiple repeated auctions in a stage, i.e., the period between two consecutive updates. For each stage, the ROI constraints need to be satisfied on average over many auctions that the bidders participate in.

Our main result is a novel dynamic mechanism achieving $\tilde{O}(T^{2/3})$ regret against a strategic automated bidder, where $T$ is the time horizon. Our dynamic mechanism adopts the explore-and-exploit scheme by first estimating the bidder's value distribution. We then offer a robust version of the single-stage revenue-optimal mechanism to extract the revenue. A cornerstone of our dynamic mechanism in the exploration phase is a novel prior-independent single-stage mechanism that is incentive-compatible for automated bidders. This mechanism forces the bidder to make a trade-off between her future gain from misreporting her values and her immediate loss, leading to a Wasserstein distance bound on the magnitude of misreporting from the automated bidder — which is precisely the form of error that our exploitation mechanisms are robust against.

## 1.2 Related work

Optimal mechanism design is a central topic in economics and game theory with many successful real-world applications, such as combinatorial auctions for FCC spectrum auctions [Cramton et al., 2006] and generalized second-price auctions for online advertising [Edelman et al., 2007]. However, almost all of these classic works make the assumption of utility-maximizing bidders. In contrast, inspired by the development of automatic bidding products in online advertising industry, a growing body of recent literature starts to focus on automated bidders. Aggarwal et al. [2019] initiate the study of mechanism design with automated bidders and raise the question of how the mechanism designed for utility-maximizing bidders performs against automated bidders. They show that neither welfare nor revenue achieved by an incentive-compatible auction can be larger than $1/2$ of the optimum. Deng et al. [2021] improve the approximation ratio to $(c+1)/(c+2)$ with boosted auctions using side information that correlates with advertisers' values. The closest related work to ours is [Balseiro et al., 2021], which studies single-stage revenue-optimal auctions for automated bidders under various information structures and bidding behavior. In particular, it examines the difference in auction design landscapes between utility-maximizing bidders and automated bidders and shows that first-best revenue can be achievable when either the the advertiser' value is public or the target ratio in the ROI constraint is public. However, their mechanisms are prior-dependent while our objective is to design prior-independent dynamic mechanisms.

Our work is also related to the literature on robust online pricing against strategic agents. No-regret policies have been developed for both the non-contextual environment [Amin et al., 2013, Drutsa, 2017, 2018, 2020] and the contextual environment [Amin et al., 2014, Deng et al., 2019, 2020, Golrezaei et al., 2019]. All these results concern impatient utility-maximizing bidders and Amin et al. [2013] show that no learning algorithm can achieve sublinear revenue loss with a patient

utility-maximizing bidder. In contrast, our work studies robust online pricing against a strategic automated bidder instead of a utility-maximizing bidder.

## 2    Preliminaries

**Dynamic mechanisms.**    Throughout this paper, we consider dynamic mechanisms that solicit a single bid from the buyer in each stage. Roughly speaking, in each stage $t$, the buyer submits their bid $b_t$, which is generally supposed to reflect their true value in stage $t$ in some way.[1] After receiving the buyer's bids, the mechanism computes the fraction $x_t$ of the item allocated to the buyer and the payment $p_t$, based on all bids that the buyer has submitted so far. Formally, let $T$ be the time horizon, i.e., the total number of stages. A dynamic mechanism $M = \{M_t\}_{t \in [T]} = \{(x_t, p_t)\}_{t \in [T]}$ is a collection of allocation-payment pairs, one for each stage $t$. For each $t$, the allocation rule $x_t : \mathbb{R}_+^t \to [0, 1]$ maps the vector of historical bids $(b_1, \ldots, b_t)$ to the fraction of the item in stage $t$ allocated to the buyer. Similarly, the payment rule $p_t : \mathbb{R}_+^t \to \mathbb{R}_+$ maps historical bids to the payment from the buyer to the seller in stage $t$.

**Buyer's incentives and behavior.**    We consider the case of automatic buyers, who are ROI constrained value maximizers. Such buyers maximize their expected value, subject to the constraint that the ratio between the value and the payment must be at least a fixed threshold in expectation. Formally, we model a value-maximizing buyer in dynamic environments in the following way. There is a value distribution $\mathcal{D} \in \Delta([0, 1])$ which is private to the buyer. In each stage $t$, the buyer's value $v_t$ is drawn from $\mathcal{D}$ independently of everything else. For simplicity, throughout the paper, we assume that $\mathcal{D}$ is non-atomic, meaning that for any $v \in [0, 1]$, $\Pr_{v' \sim \mathcal{D}}[v' = v] = 0$.[2]

There is a publicly known threshold $\tau_t \in [1, \infty)$ in each stage $t$, which specifies the minimum ROI ratio that is acceptable to the buyer in stage $t$. Note that we require $\tau_t \geq 1$ because (1) conceptually, the buyer would always want their value to be at least their payment, and (2) technically, allowing $\tau_t$ arbitrarily close to $0$ would enable the seller to extract unbounded revenue (because the buyer is willing to pay up to $v_t / \tau_t$ on average) in a single stage, in which case it is impossible to achieve low regret. Moreover, in line with the literature [Amin et al., 2013, Deng et al., 2019, Drutsa, 2017]), we assume there is a public discount factor $\lambda \in (0, 1)$, which models how impatient the buyer is.

Recall that a dynamic mechanism solicits a single bid $b_t$ from the buyer in each stage $t$. Since the mechanism decides allocation and payments based on all historical bids, the buyer may submit bids that depend on historical values in order to best respond. Thus, in general, we assume each bid $b_t : [0, 1]^t \to \mathbb{R}_+$ is a function of all historical values $v_1, \ldots, v_t$ of the buyer. Fixing a dynamic mechanism $M = \{(x_t, p_t)\}_t$, the buyer faces the following constrained optimization problem.

$$\max_{b_1, \ldots, b_T} \quad \mathbb{E}_{(v_1, \ldots, v_T) \sim \mathcal{D}^T} \left[ \sum_{t \in [T]} \lambda^t \cdot x_t \cdot v_t \right]$$

$$\text{subject to} \quad \mathbb{E}_{v_t \sim \mathcal{D}}[x_t \cdot v_t - \tau_t \cdot p_t \mid v_1, \ldots, v_{t-1}] \geq 0, \ \forall t \in [T], v_1, \ldots, v_{t-1} \in [0, 1].$$

Here, $b_t = b_t(v_1, \ldots, v_t)$, $x_t = x_t(b_1, \ldots, b_t)$, and $p_t = p_t(b_1, \ldots, b_t)$. We assume the buyer is rational, and always chooses some optimal solution to the above optimization problem as their bidding strategy. Note that here, similar to the single-stage formulation considered in previous work [Aggarwal et al., 2019, Balseiro et al., 2021, Deng et al., 2021], we only require ROI constraints to hold in expectation over $v_t$ (as opposed to for all $v_t$) in each stage. Conceptually, this is because in practice, every stage corresponds to a period of time (e.g., a day), which normally consists of a large number of auctions happening in the same environment. Therefore, it makes more sense to set a target ROI ratio for the entire time period, rather than individual transactions which are normally quite small. Technically, in-expectation constraints allow the buyer to have richer bidding strategies,

---

[1]In the Bayesian setting (i.e., with a publicly known prior), it is without loss of generality to consider incentive-compatible mechanisms where the buyer is always incentivized to submit their true value as the bid. However, the mechanism we present, being prior-independent, is not incentive-compatible. As discussed later, we only require a rough correspondence between the bids and the true values, which, as we prove, can in fact be guaranteed by the mechanism.

[2]All our results still hold without this assumption. When there are mass points in $\mathcal{D}$, the buyer's optimal bidding strategy may be random, which complicates the presentation.

and therefore, they can only make the problem of designing dynamic mechanisms harder. We also remark that one can simulate the case where the buyer submits multiple bids in each period of time, by making multiple copies of a stage with the same target ROI ratio.

**First-best revenue and seller's goal.** Our goal is to approximate the so-called first-best revenue in the long run, which is the maximum revenue possible when the buyer's values are public. Formally, suppose the seller's cost for producing a unit of the item in stage $t$ is $c_t \in [0, 1]$, and the buyer's value distribution and target ROI ratios are $\mathcal{D}$ and $\tau_t$. Then, in each stage $t$, the maximum expected revenue $\text{OPT}_t(\mathcal{D})$ the seller can achieve is given by the following optimization formulation.

$$\max_{x,p} \quad \mathbb{E}_{v \sim \mathcal{D}}[p(v) - c_t \cdot x(v)]$$
$$\text{subject to} \quad \mathbb{E}_{v \sim \mathcal{D}}[x(v) \cdot v - \tau_t \cdot p(v)] \geq 0.$$

It is known that the first-best revenue is explicitly given by

$$\text{OPT}_t(\mathcal{D}) = \mathbb{E}_{v \sim \mathcal{D}}[\max\{v/\tau_t - c_t, 0\}],$$

and that there is a single-stage incentive-compatible (meaning that it is always in the buyer's best interest to report their true value) prior-dependent mechanism that achieves single-stage revenue $\text{OPT}_t(\mathcal{D})$ [Balseiro et al., 2021]. So overall, with access to the prior distribution $\mathcal{D}$, the maximum total first-best revenue that the seller can achieve throughout the $T$ stages is

$$\text{OPT}(\mathcal{D}) = \sum_{t \in [T]} \text{OPT}_t(\mathcal{D}).$$

In this paper, we assume that the seller does not have access to the prior distribution $\mathcal{D}$, and the goal is to achieve sublinear regret against the first-best revenue $\text{OPT}(\mathcal{D})$ for all $\mathcal{D}$ with a single dynamic mechanism. Formally, we aim to find a mechanism $M = \{(x_t, p_t)\}_{t \in [T]}$, subject to the condition that for all $\mathcal{D}$, there exists a bidding strategy satisfying the buyer's ROI constraints,[3] and moreover,

$$\sup_{\mathcal{D} \in \Delta(V)} \left( \text{OPT}(\mathcal{D}) - \sum_{t \in [T]} \left( p_t(b_1^{\mathcal{D},M}, \ldots, b_t^{\mathcal{D},M}) - x_t(b_1^{\mathcal{D},M}, \ldots, b_t^{\mathcal{D},M}) \cdot c_t \right) \right) = o(T),$$

where $b_t^{\mathcal{D},M} = b_t^{\mathcal{D},M}(v_1, \ldots, v_t)$ is the buyer's best response to mechanism $M$ when the value distribution is $\mathcal{D}$, as defined above.

## 3 The Mechanism

In this section, we present the overall mechanism and discuss key high-level ideas therein. The full mechanism is given in Figure 1. Similar to prior work [Amin et al., 2013], our mechanism has an exploration phase (step 2) followed by an exploitation phase (step 4). The idea of the exploration phase is to obtain an estimate $\hat{\mathcal{D}}$ of the buyer's value distribution $\mathcal{D}$, by offering prior-independent auctions that are *strictly*[4] single-stage incentive-compatible, in the sense that restricted to the current stage, the more the buyer misreports, the worse their value will be (but misreporting may still lead to future gain). In each of these auctions, the seller can potentially suffer significant loss. However, by keeping the exploration phase short, the total loss from this phase can be bounded. Here, we face several technical challenges introduced by value-maximizing buyers, including:

- Single-stage incentive-compatible mechanisms for ROI constrained buyers have substantially different structures than those for quasilinear utility-maximizing buyers that are traditionally considered. In fact, even the existence of a strictly single-stage incentive-compatible prior-independent mechanism for ROI constrained buyers is nontrivial, let alone finding one explicitly that specifically suits our purposes.

---

[3]For example, a mechanism where $p_t(0) = 0$ for all $t$ would satisfy this condition. In particular, $b_t = 0$ for all $t$ is always a feasible bidding strategy under such mechanisms.

[4]Note that weak incentive-compatibility, which guarantees that the buyer is indifferent between truth-telling and lying, does not suffice for our purposes. In particular, the buyer may still have strict incentive to misreport in arbitrary ways in response to such mechanisms, as long as misreporting leads to any positive future gain (no matter how small it is or how much it is discounted).

1. Let $\varepsilon = T^{-1/3}$, $T_1 = \Theta(\log T/\varepsilon^2)$, and $T_2 = \Theta\left(\frac{\log((1-\lambda)\cdot\varepsilon^2)}{\log\lambda}\right)$, where $T$ is the time horizon and $\lambda$ is the buyer's discount factor.

2. For each $t = 1, \ldots, T_1 + T_2$, solicit a bid $b_t \in \mathbb{R}_+$ from the buyer, allocate $x_t = \min\{b_t, 1\}$ of the item to the buyer and charge payment $p_t = x_t \cdot \frac{\min\{b_t, 1\}}{\tau_t}$.

3. Let $\hat{\mathcal{D}}$ be the distribution induced by $\{\min\{b_t, 1\}\}_{t\in[T_1]}$, i.e., for each $v \in [0, 1]$,
$$\hat{\mathcal{D}}(v) = \Pr_{v'\sim\hat{\mathcal{D}}}[v' = v] = \frac{\sum_{t\in[T_1]}\mathbb{I}[\min\{b_t, 1\} = v]}{T_1}.$$

4. For each $t = T_1 + T_2 + 1, \ldots, T$, let
$$q_t = \mathbb{E}_{v\sim\hat{\mathcal{D}}}[v/\tau_t \mid v/\tau_t \geq c_t] - \frac{\varepsilon}{\Pr_{v\sim\hat{\mathcal{D}}}[v/\tau_t \geq c_t]}.$$

Solicit a bid $b_t \in \mathbb{R}_+$. If $\frac{b_t}{\tau_t} \geq c_t$, then allocate $x_t = 1$ of the item and charge $p_t = \max\{q_t, c_t\}$. Otherwise, allocate $x_t = 0$ and charge $p_t = 0$.

Figure 1: A prior-indepedent no-regret mechanism for a value-maximizing buyer.

- In conjunction with the strictly single-stage incentive-compatible mechanism, we need to find the right way of measuring the magnitude of misreport, which must be manageable given the mechanism we choose, and at the same time ensure that our estimate $\hat{\mathcal{D}}$ is sufficiently close to $\mathcal{D}$ in a way that can be exploited in the later phase.

To tackle the above challenges, we use an exploration mechanism that is single-stage incentive-compatible in a very specific way. Intuitively, single-stage incentive-compatibility for ROI constrained buyers roughly means the buyer cannot gain by "arbitraging" between value and ROI slackness. Our exploration mechanism is designed such that whenever the buyer tries to arbitrage by misreporting, they suffer an immediate loss in the sum of value and ROI slackness that is precisely equal to the squared magnitude of deviation from their true value. Dynamic incentive-compatibility then allows us to bound the expected squared deviation in any single stage in the exploration phase.

When computing our estimate $\hat{\mathcal{D}}$, we only use the first $T_1$ bids collected in the exploration phase. This is because the last $T_2$ bids are chronologically too close to the exploitation phase. As a result, the buyer's interest in future gain is not sufficiently discounted when making these bids so that they are potentially more misleading than the first $T_1$ bids in the exploration phase. For this reason, we simply ignore the last $T_2$ bids to avoid introducing error in our estimate that cannot be effectively controlled.

Finally, in the exploitation phase, we switch to running nearly revenue-optimal auctions based on our estimate $\hat{\mathcal{D}}$ of the value distribution $\mathcal{D}$ obtained from the exploration phase. Incentive-compatible revenue-optimal mechanisms for ROI constrained buyers have been characterized by Balseiro et al. [2021]. However, such optimal mechanisms require knowing the exact value distribution $\mathcal{D}$, and it is not immediately clear whether an estimate suffices for approximate optimality. So here, the main challenge is to adapt previously known optimal mechanisms into a robust form that tolerates the specific form of error introduced in the exploration phase.

We remark that the mechanism does not need to know the time horizon $T$ beforehand: applying the standard doubling trick allows the mechanism to achieve the same regret bound for any unknown $T$ [Deng et al., 2019, 2020, Golrezaei et al., 2019]. Also, the mechanism does not need to know $\tau_t$ and $c_t$ before each stage $t$.

## 4 Analysis of the Mechanism

In this section, we present an analysis of our mechanism and establish the regret bound. We first consider the exploration phase in Section 4.1, aiming to bound the difference between the true value

distribution $\mathcal{D}$ and our estimate $\hat{\mathcal{D}}$. Then in Section 4.2, we turn to the exploitation phase, and show that the mechanisms used there are robust against the specific form of estimation error introduced in the exploration phase. Finally, we combine everything into a regret bound for our mechanism.

## 4.1 The Exploration Phase

In this section, we show that the exploration phase yields an accurate estimate of the buyer's value distribution with high probability, whenever the buyer adopts an optimal bidding strategy. The first key step is to prove the strict single-stage incentive-compatibility of the mechanism used in each stage $t$ in the exploration phase (henceforth the exploration mechanism in stage $t$). In fact, each exploration mechanism belongs to a larger class of nontrivial prior-independent mechanisms that are single-stage incentive-compatible for ROI constrained buyers. Before zooming into exploration mechanisms, we first prove the single-stage incentive-compatibility of this larger class of mechanisms, which may be of independent interest.

**Proposition 1.** *Fix a target ROI ratio $\tau$. For any $\alpha \in \mathbb{R}_+$, let $(x_\alpha, p_\alpha)$ be such that for any $b \in \mathbb{R}_+$,*

$$x_\alpha(b) = (\min\{b, 1\})^\alpha \quad and \quad p_\alpha(b) = x_\alpha(b) \cdot \frac{\min\{b, 1\}}{\tau}.$$

*Then fixing any $\alpha \in \mathbb{R}_+$, for all value distribution $\mathcal{D}$, $(x_\alpha, p_\alpha)$ is single-stage incentive-compatible, i.e., for any bidding strategy $b : [0, 1] \to \mathbb{R}_+$ satisfying*

$$\mathop{\mathbb{E}}_{v \sim \mathcal{D}}[x_\alpha(b(v)) \cdot v - \tau \cdot p_\alpha(b(v))] \geq 0,$$

*we have*

$$\mathop{\mathbb{E}}_{v \sim \mathcal{D}}[x_\alpha(b(v)) \cdot v] \leq \mathop{\mathbb{E}}_{v \sim \mathcal{D}}[x_\alpha(v) \cdot v].$$

We defer the proof of the proposition, as well as all other missing proofs, to the appendix. In principle, for any $\alpha > 0$, $(x_\alpha, p_\alpha)$ is strictly single-stage incentive-compatible. Out of all these mechanisms, we choose the one corresponding to $\alpha = 1$ as our exploration mechanisms (instantiated with different $\tau_t$ in different stages), because it enables a way of bounding the magnitude of misreporting that couples nicely with other components of our overall mechanism. Formally, the exploration mechanism in each stage $t$ has the following key property.

**Lemma 1.** *Let $(x_t, p_t)$ be the exploration mechanism in stage $t$. For any value distribution $\mathcal{D}$ and bidding strategy $b : [0, 1] \to [0, 1]$, let*

$$\mathrm{Val}(\mathcal{D}, b) := \mathop{\mathbb{E}}_{v \sim \mathcal{D}}[x(b(v)) \cdot v] \quad and \quad \mathrm{ROI}(\mathcal{D}, b) := \mathop{\mathbb{E}}_{v \sim \mathcal{D}}[x(b(v)) \cdot v - \tau_t \cdot p(b(v))].$$

*Moreover, let $\mathrm{id} : [0, 1] \to [0, 1]$ be the truthful bidding strategy, where $\mathrm{id}(v) = v$ for all $v \in [0, 1]$. Then for any bidding strategy $b : [0, 1] \to [0, 1]$,*

$$\mathrm{Val}(\mathcal{D}, b) + \mathrm{ROI}(\mathcal{D}, b) = \mathrm{Val}(\mathcal{D}, \mathrm{id}) - \mathop{\mathbb{E}}_{v \sim \mathcal{D}}[(v - b(v))^2].$$

*Proof.* First fix any $v \in [0, 1]$ and $v' = b(v)$. We have $x(v') \cdot v - x(v) \cdot v = v \cdot (v' - v)$, and moreover, $(x(v') \cdot v - \tau_t \cdot p(v')) - (x(v) \cdot v - \tau_t \cdot p(v)) = v'(v - v')$. As a result, for any $\mathcal{D}$ and $b$,

$$\begin{aligned}
&\big(\mathrm{Val}(\mathcal{D}, b) + \mathrm{ROI}(\mathcal{D}, b)\big) - \big(\mathrm{Val}(\mathcal{D}, \mathrm{id}) + \mathrm{ROI}(\mathcal{D}, \mathrm{id})\big) \\
&= \mathop{\mathbb{E}}_{v \sim \mathcal{D}}\left[\big(x(b(v)) \cdot v - x(v) \cdot v\big) + \big((x(b(v)) \cdot v - \tau_t \cdot p(b(v))) - (x(v) \cdot v - \tau_t \cdot p(v))\big)\right] \\
&= \mathop{\mathbb{E}}_{v \sim \mathcal{D}}\left[v \cdot (b(v) - v) + b(v) \cdot (v - b(v))\right] = \mathop{\mathbb{E}}_{v \sim \mathcal{D}}\left[-(v - b(v))^2\right].
\end{aligned}$$

Rearranging terms and plugging in $\mathrm{ROI}(\mathcal{D}, \mathrm{id}) = 0$, we obtain precisely the claim to be proved. $\square$

The above lemma states that under the exploration mechanism in each stage $t$, any bidding strategy that deviates from truthful reporting suffers a loss in the sum of expected value (from the fraction of the item received) and ROI slackness. And moreover, this loss is equal to the expected squared magnitude of deviation. This provides a way of bounding the magnitude of deviation in each stage in the exploration phase because of the following reasons. First, since the buyer is ROI constrained, in each stage they must have nonnegative ROI slackness, and as a result, their immediate value loss

is at least as large as the squared magnitude of deviation in the current stage. Moreover, note that the exploration phase has completely no dependence on historical bids, so the only motivation for misreporting in the exploration phase comes from the exploitation phase, which is far away in the future as we only use the first $T_1$ bids in the estimation. Since the buyer is less patient than the seller (i.e., they have a discount factor $\lambda < 1$), any potential future gain is substantially discounted. As a result, the motivation for deviation becomes sufficiently small, such that the buyer is only willing to sacrifice a small amount of immediate value in exchange for that. Thus, by Lemma 1, we know that the expected squared magnitude of deviation in each exploration phase auction is sufficiently small. This is captured by the following lemma.

**Lemma 2.** *For any value distribution $\mathcal{D}$, optimal bidding strategy $\{b_t\}$, $t \in [T_1]$, and historical values $v_1, \ldots, v_{t-1}$ before stage $t$,*

$$\mathop{\mathbb{E}}_{v_t \sim \mathcal{D}}[(v_t - b_t)^2 \mid v_1, \ldots, v_{t-1}] \leq \frac{\varepsilon^2}{16},$$

*and as a corollary,*

$$\mathop{\mathbb{E}}_{v_t \sim \mathcal{D}}[|v_t - b_t| \mid v_1, \ldots, v_{t-1}] \leq \frac{\varepsilon}{4},$$

*where $\varepsilon$ is the error parameter defined in Figure 1.*

The above lemma controls one source (i.e., misreporting) of error incurred in the exploration phase. In particular, the form of error that we care about is in the second part of the lemma, which provides an upper bound on the expected absolute value of deviation in any exploration mechanism. This measure of error is closely related to the notion of the $\ell_1$ Wasserstein distance $W_1$ (also known as the earth mover's distance) in $\mathbb{R}$, defined as follows.

**Definition 1.** For two probability distributions $\mathcal{D}_1$ and $\mathcal{D}_2$ over $\mathbb{R}$,

$$W_1(\mathcal{D}_1, \mathcal{D}_2) := \inf_{\mathcal{D}_j \in \Gamma(\mathcal{D}_1, \mathcal{D}_2)} \left( \mathop{\mathbb{E}}_{(x,y) \sim \mathcal{D}_j}[|x - y|] \right),$$

where $\Gamma(\mathcal{D}_1, \mathcal{D}_2)$ is the family of distributions over $\mathbb{R}^2$ with marginals $\mathcal{D}_1$ and $\mathcal{D}_2$ in the two dimensions respectively.

Intuitively, each $\mathcal{D}_j \in \Gamma(\mathcal{D}_1, \mathcal{D}_2)$ corresponds to a matching between $\mathcal{D}_1$ and $\mathcal{D}_2$, and $W_1(\mathcal{D}_1, \mathcal{D}_2)$ is the minimum total cost of matching $\mathcal{D}_1$ to $\mathcal{D}_2$. In fact, it is easy to show that Lemma 2 implies the following bound.

**Corollary 1.** *Let $b_t(\mathcal{D})$ be the distribution of $b_t(v_t)$ (given $v_1, \ldots, v_{t-1}$) where $v_t \sim \mathcal{D}$. Then*

$$W_1(\mathcal{D}, b_t(\mathcal{D})) \leq \mathop{\mathbb{E}}_{v_t \sim \mathcal{D}}[|v_t - b_t(v_t)|] \leq \frac{\varepsilon}{4}.$$

This is simply because $b_t$ explicitly specifies a matching between $\mathcal{D}$ and $b_t(\mathcal{D})$, whose cost is upper bounded by $\varepsilon/4$. In light of this, our goal in the rest of Section 4.1 is to upper bound $W_1(\mathcal{D}, \hat{\mathcal{D}})$. We will later see in Section 4.2 that this is precisely the kind of guarantee that our exploitation mechanisms need to achieve approximate optimality.

Below we handle the other source of error, i.e., the discrepency between the buyer's value distribution $\mathcal{D}$ and the empirical value distribution (henceforth $\bar{\mathcal{D}}$) without misreporting, induced by $\{v_t\}_{t \in [T_1]}$. Since $\{v_t\}_{t \in [T_1]}$ are i.i.d. samples from $\mathcal{D}$, we can apply standard results for the convergence of empirical processes in the $W_1$ distance to obtain the following bound.

**Lemma 3.** *Let $\bar{\mathcal{D}}$ be the empirical distribution induced by $v_1, \ldots, v_{T_1}$. With probability at least $1 - 1/(2T)$ over $v_1, \ldots, v_{T_1}$, $W_1(\mathcal{D}, \bar{\mathcal{D}}) \leq \frac{\varepsilon}{2}$.*

With Lemma 3 proved, we can now proceed to the final step of bounding $W_1(\mathcal{D}, \hat{\mathcal{D}})$, i.e., combining the error from misreporting and that from $W_1(\mathcal{D}, \bar{\mathcal{D}})$. This may appear straightforward: the error from misreporting is on top of $W_1(\mathcal{D}, \bar{\mathcal{D}})$, so we only need to add the bounds from Corollary 1 and Lemma 3 together. However, to achieve low regret, we need $W_1(\mathcal{D}, \hat{\mathcal{D}})$ to be small with high probability, but Corollary 1 only provides an in-expectation bound. Moreover, since the buyer may use an adaptive bidding strategy (i.e., $b_t$ may depend on $b_1, \ldots, b_{t-1}$), we cannot apply standard

concentration bounds for independent variables to obtain high-probability bounds. Therefore, instead of applying Corollary 1, we observe that the sequence

$$X_t = \frac{1}{T_1} \sum_{t' \leq t} \left( |b_{t'} - v_{t'}| - \frac{\varepsilon}{4} \right)$$

form a submartingale. As a result, we can apply one-sided concentration inequalities for submartingales to obtain a high-probability upper bound on

$$X_{T_1} = \frac{1}{T_1} \sum_{t \in [T_1]} \left( |b_t - v_t| - \frac{\varepsilon}{4} \right).$$

The above observation gives us the following lemma.

**Lemma 4.** *With probability at least* $1 - 1/(2T)$ *over* $v_1, \ldots, v_{T_1}$,

$$W_1(\bar{\mathcal{D}}, \hat{\mathcal{D}}) \leq \frac{1}{T_1} \sum_{t \in [T_1]} |b_t - v_t| \leq \frac{\varepsilon}{2}.$$

Combining Lemmas 3 and 4 , we immediately obtain the following high-probability bound on $W_1(\mathcal{D}, \hat{\mathcal{D}})$, which is essentially the only property of the exploration phase that we need to establish the overall regret bound.

**Lemma 5.** *With probability at least* $1 - 1/T$, $W_1(\mathcal{D}, \hat{\mathcal{D}}) \leq \varepsilon$.

### 4.2 The Exploitation Phase

In this section, we show that when the exploration phase succeeds, i.e., when $W_1(\mathcal{D}, \hat{\mathcal{D}}) \leq \varepsilon$, the expected revenue by the mechanism in each stage $t$ in the exploitation phase (henceforth the exploitation mechanism in stage $t$) is at least $\mathrm{OPT}_t(\mathcal{D}) - \varepsilon$. Since the bid submitted in any of these stages cannot affect the exploitation mechanism in any other stage, we can decouple the exploitation stages and look at each of them individually.

We first characterize the buyer's optimal bidding strategy in response to the exploitation mechanism in each stage $t$. First observe that these exploitation mechanisms have a simple structure: the exploitation mechanism in stage $t$ is essentially a take-it-or-leave-it offer, where the buyer can either take the item and pay $q_t$, or leave and pay nothing. In particular, the specific value of the bid $b_t$ does not affect how much the buyer pays when they get the item.

Under the traditional assumption that the buyer maximizes their utility, the best response to such mechanisms is extremely simple: take the item if and only if the buyer's value $v_t$ at time $t$ is at least the posted price $q_t$. The optimal bidding strategy is less straightforward with value-maximizing buyers who maximize value subject to ROI constraints. However, they still have the intuitve structure that the buyer should take the item whenever their value is higher than a certain threshold. This is captured by the following claim.

**Lemma 6.** *Suppose a value-maximizing buyer with target ROI ratio* $\tau$ *has value distribution* $\mathcal{D}$. *Then the optimal bidding strategy* $b : [0, 1] \rightarrow [0, 1]$ *against a take-it-or-leave-it offer at price* $q \in (0, 1)$ *is given by*

$$b(v) = \begin{cases} 1, & \text{if } v \geq \theta \\ 0, & \text{otherwise,} \end{cases}$$

*where*

$$\theta = \inf\{\theta' \mid \mathbb{E}_{v \sim \mathcal{D}}[v \mid v \geq \theta'] \geq \tau \cdot q\}.$$

We remark that the above optimal bidding strategy is essentially unique, in the sense that the mappings from the value $v$ to the allocation $x$ induced by any two optimal bidding strategies can differ only on a zero-measure set with respect to $\mathcal{D}$, which makes no difference for our purposes. Henceforth, in the rest of the paper, without loss of generality we assume that the bidding strategy given in Lemma 6 is the unique optimal bidding strategy.

With the optimal bidding strategy characterized, we can now bound the revenue collected in the exploitation mechanism in each stage $t$ and compare that to $\mathrm{OPT}_t(\mathcal{D})$. We first prove that when the

exploration phase succeeds, in each exploitation stage, the optimal revenue that can be extracted from the buyer when the value distribution is $\mathcal{D}$ cannot be much larger than that when the value distribution is $\hat{\mathcal{D}}$. Formally, we have the following lemma.

**Lemma 7.** *When $W_1(\mathcal{D}, \hat{\mathcal{D}}) \leq \varepsilon$, for all $t \in \{T_1 + T_2 + 1, \ldots, T\}$,*

$$\mathrm{OPT}_t(\mathcal{D}) \leq \mathrm{OPT}_t(\hat{\mathcal{D}}) + \varepsilon.$$

The next step, which captures the essence of our exploitation mechanisms, is to show that, conditioning on the success of the exploration phase, when the buyer (with actual value distribution $\mathcal{D}$) uses the optimal bidding strategy in the exploitation mechanism in stage $t$, the revenue collected is not much smaller than $\mathrm{OPT}_t(\hat{\mathcal{D}})$. In other words, the exploitation mechanism in stage $t$ can tolerate an estimation error of up to $\varepsilon$ in the $W_1$ distance, sacrificing only $\varepsilon$ in the revenue. Intuitively, this is because by the definition of the $W_1$ distance, we can find a matching $\mathcal{D}_j \in \Gamma(\mathcal{D}, \hat{\mathcal{D}})$ between $\mathcal{D}$ and $\hat{\mathcal{D}}$, such that the following is a feasible bidding strategy for a buyer with distribution $\mathcal{D}$: for $(v, \hat{v}) \sim \mathcal{D}_j$, where $v$ is the realized value of the buyer and $\hat{v}$ is the value that $v$ matches to under $\mathcal{D}_j$, buy the item if and only if a buyer with distribution $\hat{\mathcal{D}}$ would buy the item when their value is $\hat{v}$ in response to a slightly higher price. This means the probability that a buyer with distribution $\mathcal{D}$ buys the item is at least the probability that a buyer with distribution $\hat{\mathcal{D}}$ does when the price is slightly higher. Then, one can show that under the optimal bidding strategy, a buyer with distribution $\mathcal{D}$ buys the item with at least the same probability, so the revenue loss is solely from the difference between the two prices, which is by construction small. This intuition is captured by the following lemma.

**Lemma 8.** *When $W_1(\mathcal{D}, \hat{\mathcal{D}}) \leq \varepsilon$, for all $t \in \{T_1 + T_2 + 1, \ldots, T\}$, the expected revenue of the exploitation mechanism in stage $t$ satisfies*

$$\mathrm{Rev}_t(\mathcal{D}, \hat{\mathcal{D}}) := \max\{(q_t - c_t), 0\} \cdot \Pr_{v \sim \mathcal{D}}[v \geq \theta_t] \geq \mathrm{OPT}_t(\hat{\mathcal{D}}) - \varepsilon,$$

*where $\theta_t$ is the threshold of the optimal bidding strategy as given in Lemma 6, i.e.,*

$$\theta_t = \inf\{\theta \in [0, 1] \mid \mathbb{E}_{v \sim \mathcal{D}}[v \mid v \geq \theta] \geq \tau_t \cdot q_t\}.$$

Finally we combine Lemmas 7 and 8 to obtain the following conditional revenue bound, which is essentially the only property of exploitation mechanisms that we need.

**Lemma 9.** *When $W_1(\mathcal{D}, \hat{\mathcal{D}}) \leq \varepsilon$, for all $t \in \{T_1 + T_2 + 1, \ldots, T\}$, the expected revenue of the exploitation mechanism in stage $t$ satisfies*

$$\mathrm{Rev}_t(\mathcal{D}, \hat{\mathcal{D}}) \geq \mathrm{OPT}_t(\mathcal{D}) - 2\varepsilon.$$

### 4.3 Putting Everything Together

Finally, we are ready to put everything together and prove the regret bound of our mechanism. The following theorem states that our mechanism not only guarantees low regret in expectation, but also in fact collects total revenue close to the first-best benchmark with high probability, over the the randomness in the buyer's values.

**Theorem 1.** *With probability at least $1 - O(1/T)$, our mechanism achieves total revenue at least*

$$\mathrm{OPT}(\mathcal{D}) - O(T^{2/3} \log T).$$

## 5 Conclusion

In this paper, we develop a prior-independent dynamic mechanism achieving low regret against a value-maximizing buyer. Along the way, we propose a novel prior-independent single-stage mechanism that is incentive-compatible for value-maximizing buyers. An intriguing question to consider in the future is to extend the results to an environment with multiple value-maximizing buyers; however, tackling the interactions between multiple buyers seems to be a non-trivial and challenging task. Another open question is whether the $\tilde{O}(T^{2/3})$ bound is (nearly) optimal — we suspect the bound cannot be significantly improved restricted to exploration-exploitation mechanisms,

given that similar lower bounds have been established in related problems in prior work [Babaioff et al., 2014, Devanur and Kakade, 2009]. In principle, one could also consider the case where the target ROI ratio $\tau_t$ is also private and needs to be reported. However, people do not currently have a good understanding of this case even in static environments, and as a result, it is not even clear what the benchmark should be for this case in dynamic environments. Other variants of the problem, e.g., where there is a "cumulative" ROI constraint over all stages (as opposed to one target ROI ratio for each stage), may also be of potential interest. To zoom out, it is very interesting to establish a better understanding about the power and the limitations of dynamic mechanisms against value-maximizing buyers.

## Funding Transparency Statement

Hanrui Zhang was supported by NSF grant IIS-1814056.

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
