---

1. Let $\varepsilon = T^{-1/3}$, $T_1 = \Theta(\log T/\varepsilon^2)$, and $T_2 = \Theta\left(\frac{\log((1-\lambda)\cdot\varepsilon^2)}{\log \lambda}\right)$, where $T$ is the time horizon and $\lambda$ is the buyer's discount factor.

2. For each $t = 1, \dots, T_1 + T_2$, solicit a bid $b_t \in \mathbb{R}_+$ from the buyer, allocate $x_t = \min\{b_t, 1\}$ of the item to the buyer and charge payment $p_t = x_t \cdot \frac{\min\{b_t, 1\}}{\tau_t}$.

3. Let $\hat{\mathcal{D}}$ be the distribution induced by $\{\min\{b_t, 1\}\}_{t \in [T_1]}$, i.e., for each $v \in [0, 1]$,

$$\hat{\mathcal{D}}(v) = \Pr_{v' \sim \hat{\mathcal{D}}}[v' = v] = \frac{\sum_{t \in [T_1]} \mathbb{I}[\min\{b_t, 1\} = v]}{T_1}.$$

4. For each $t = T_1 + T_2 + 1, \dots, T$, let

$$q_t = \mathop{\mathbb{E}}_{v \sim \hat{\mathcal{D}}}[v/\tau_t \mid v/\tau_t \ge c_t] - \frac{\varepsilon}{\Pr_{v \sim \hat{\mathcal{D}}}[v/\tau_t \ge c_t]}.$$

Solicit a bid $b_t \in \mathbb{R}_+$. If $\frac{b_t}{\tau_t} \ge c_t$, then allocate $x_t = 1$ of the item and charge $p_t = \max\{q_t, c_t\}$. Otherwise, allocate $x_t = 0$ and charge $p_t = 0$.

---

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

$$\text{Val}(\mathcal{D}, b) + \text{ROI}(\mathcal{D}, b) = \text{Val}(\mathcal{D}, \text{id}) - \mathbb{E}_{v \sim \mathcal{D}}[(v - b(v))^2].$$

*Proof.* First fix any $v \in [0, 1]$ and $v' = b(v)$. We have $x(v') \cdot v - x(v) \cdot v = v \cdot (v' - v)$, and moreover, $(x(v') \cdot v - \tau_t \cdot p(v')) - (x(v) \cdot v - \tau_t \cdot p(v)) = v'(v - v')$. As a result, for any $\mathcal{D}$ and $b$,

$$\big(\text{Val}(\mathcal{D}, b) + \text{ROI}(\mathcal{D}, b)\big) - \big(\text{Val}(\mathcal{D}, \text{id}) + \text{ROI}(\mathcal{D}, \text{id})\big)$$

$$= \mathbb{E}_{v \sim \mathcal{D}} \Big[ \big(x(b(v)) \cdot v - x(v) \cdot v\big) + \Big(\big(x(b(v)) \cdot v - \tau_t \cdot p(b(v))\big) - \big(x(v) \cdot v - \tau_t \cdot p(v)\big)\Big)\Big]$$

$$= \mathbb{E}_{v \sim \mathcal{D}} \big[v \cdot (b(v) - v) + b(v) \cdot (v - b(v))\big] = \mathbb{E}_{v \sim \mathcal{D}} \big[-(v - b(v))^2\big].$$

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

# A   Omitted Proofs

*Proof of Proposition 1.* The case of $\alpha = 0$ is trivial. For $\alpha > 0$, we use the following characterization (rephrased for our purposes) of single-stage incentive-compatible mechanisms for ROI constrained buyers given in [Balseiro et al., 2021]: under mild assumptions (Assumption 3.1 in [Balseiro et al., 2021]) that are satisfied by $(x_\alpha, p_\alpha)$ for any $\alpha \in (0, \infty)$, a mechanism $(x, p)$ is incentive-compatible iff there exists $\gamma \geq 0$ such that

1. For all $v, v' \in [0, 1]$,

$$(1 + \gamma) \cdot v \cdot x(v) - \gamma \cdot \tau \cdot p(v) \geq (1 + \gamma) \cdot v \cdot x(v') - \gamma \cdot \tau \cdot p(v').$$

2. Both of the following hold, and at least one of them achieves equality:

$$\mathbb{E}_{v \sim \mathcal{D}}[v \cdot x(v) - \tau \cdot p(v)] \geq 0 \quad \text{and} \quad \gamma \geq 0.$$

For the class of mechanisms we consider, we always have

$$p_\alpha(v) = x_\alpha(v) \cdot \frac{v}{\tau},$$

so condition 2 always holds, and condition 1 simplifies to: for all $v, v'$,

$$v \cdot x_\alpha(v) \geq (1 + \gamma) \cdot v \cdot x_\alpha(v') - \gamma \cdot v' \cdot x_\alpha(v') \iff v \cdot (x_\alpha(v) - x_\alpha(v')) \geq \gamma \cdot x_\alpha(v') \cdot (v - v').$$

Below we show that the above is satisfied by $x_\alpha$ for any $\alpha \in (0, 1]$. We choose

$$\gamma = \alpha = \frac{v \cdot \dot{x}_\alpha(v)}{x_\alpha(v)},$$

for all $v \in (0, 1]$. First suppose $v > v'$. In such cases, the above condition is equivalent to

$$\frac{v \cdot (x_\alpha(v) - x_\alpha(v'))}{x_\alpha(v') \cdot (v - v')} \geq \gamma.$$

By the mean value theorem, for some $v'' \in (v', v)$,

$$\frac{v \cdot (x_\alpha(v) - x_\alpha(v'))}{x_\alpha(v') \cdot (v - v')} = \frac{v \cdot \dot{x}_\alpha(v'')}{x_\alpha(v')}.$$

Since $x_\alpha$ is non-decreasing, we have

$$\frac{v \cdot \dot{x}_\alpha(v'')}{x_\alpha(v')} \geq \frac{v'' \cdot \dot{x}_\alpha(v'')}{x_\alpha(v'')} = \alpha = \gamma.$$

Now suppose $v < v'$. In such cases, the condition is equivalent to

$$\frac{v \cdot (x_\alpha(v) - x_\alpha(v'))}{x_\alpha(v') \cdot (v - v')} \leq \gamma.$$

Again, by the mean value theorem and the fact that $x_\alpha$ is non-decreasing, for some $v'' \in (v, v')$,

$$\frac{v \cdot (x_\alpha(v) - x_\alpha(v'))}{x_\alpha(v') \cdot (v - v')} = \frac{v \cdot \dot{x}_\alpha(v'')}{x_\alpha(v')} \leq \frac{v'' \cdot \dot{x}_\alpha(v'')}{x_\alpha(v'')} = \alpha = \gamma.$$

This concludes the proof. $\qquad\square$

*Proof of Lemma 2.* Fix some $t$, and consider the incentives of the buyer in stage $t$. Note that an optimal bidding strategy must be conditionally optimal given any historical values $v_1, \ldots, v_{t-1}$ before stage $t$. Since the exploration phase of our mechanism are both history-independent, all potential gain of misreporting in stage $t$ comes from the exploitation phase. In each stage $t'$ in the exploitation phase, the immediate gain of the buyer from misreporting in stage $t$ is at most 1, and the contribution to the value in stage $t$ is at most $\lambda^{t'-t}$ due to discounting. So, overall, the maximum possible future gain from misreporting in stage $t$ is at most

$$\sum_{t+T_2 \leq t' \leq T} \lambda^{t'-t} \leq \lambda^{T_2} \cdot \sum_{i \in [T]} \lambda^i \leq \frac{\lambda^{T_2}}{1-\lambda}.$$

Now in order for $b_t$ to be conditionally optimal, the future gain must outweigh the immediate loss, so by Lemma 1,

$$\mathop{\mathbb{E}}_{v_t \sim \mathcal{D}}[(v_t - b_t)^2 \mid v_1, \ldots, v_{t-1}] = \text{Val}(\mathcal{D}, \text{id}) - \text{Val}(\mathcal{D}, b_t) - \text{ROI}(\mathcal{D}, b_t)$$

$$\leq \text{Val}(\mathcal{D}, \text{id}) - \text{Val}(\mathcal{D}, b_t) \qquad (\text{ROI}(\mathcal{D}, b_t) \geq 0)$$

$$\leq \frac{\lambda^{T_2}}{1-\lambda} \qquad (\text{optimality of } b_t)$$

$$\leq \frac{\varepsilon^2}{16}. \qquad (\text{choice of } T_2)$$

This concludes the proof of the first part. For the corollary, by Jensen's inequality,

$$\mathop{\mathbb{E}}_{v_t \sim \mathcal{D}}[|v_t - b_t| \mid v_1, \ldots, v_{t-1}] \leq \sqrt{\mathop{\mathbb{E}}_{v_t \sim \mathcal{D}}[(v_t - b_t)^2 \mid v_1, \ldots, v_{t-1}]} \leq \frac{\varepsilon}{4}. \qquad\square$$

*Proof of Lemma 3.* We use the result in [Boissard and Le Gouic, 2014] to upper bound $\mathbb{E}[W_1(\mathcal{D}, \bar{\mathcal{D}})]$, and then apply McDiarmid's inequality to translate that into a high-probability bound. First, by Theorem 1.1 of [Boissard and Le Gouic, 2014], setting $p = 1$, $n = T_1$, $\mu = \mathcal{D}$, and $t = n^{-1/2}$, we have

$$\mathbb{E}[W_1(\mathcal{D}, \hat{\mathcal{D}})] \leq 64 T_1^{-1/2} \leq \frac{\varepsilon}{4}.$$

Now observe that by the definition of $W_1(\mathcal{D}, \bar{\mathcal{D}})$, the dependence of $W_1(\mathcal{D}, \hat{\mathcal{D}}) = W_1(v_1, \ldots, v_{T_1})$ on each $v_t$ is

$$\max_{v_1, \ldots, v_{T_1}, v_t'} |W_1(v_1, \ldots, v_t, \ldots, v_{T_1}) - W_1(v_1, \ldots, v_t', \ldots, v_{T_1})| \leq \frac{1}{T_1}.$$

Moreover, $\{v_t\}_{t \in [T_1]}$ are independent. So, by McDiarmid's inequality, we have

$$\Pr[W_1(\mathcal{D}, \bar{\mathcal{D}}) \leq \mathbb{E}[W_1(\mathcal{D}, \bar{\mathcal{D}})] + \varepsilon/4] \leq \exp\left(-\frac{\varepsilon^2 \cdot T_1}{8}\right) \leq \frac{1}{2T}.$$

The claim follows. $\qquad\square$

*Proof of Lemma 4.* For each $0 \leq t \leq T_1$, let

$$X_t = \frac{1}{T_1} \sum_{t' < t} \left(|b_{t'} - v_{t'}| - \frac{\varepsilon}{4}\right).$$

Observe that $\{X_t\}$ form a supermartingale. In particular, for any $0 \le t < T_1$, by Lemma 2,

$$\mathbb{E}[X_{t+1} \mid X_0, \dots, X_t] = X_t + \frac{1}{T_1} \cdot \mathbb{E}\left[|b_t - v_t| - \frac{\varepsilon}{4} \mid X_0, \dots, X_t\right] \le X_t.$$

And moreover, since $0 \le |b_t - v_t| \le 1$,

$$\mathrm{Var}[X_{t+1} \mid X_0, \dots, X_t] \le \frac{1}{4T_1^2} \quad \text{and} \quad X_{t+1} - \mathbb{E}[X_{t+1} \mid X_0, \dots X_t] \le \frac{1}{T_1}.$$

So applying Theorem 27 of [Chung and Lu, 2006] (with $M = 1/T_1$, $a_i = \phi_i = 0$ and $\sigma_i^2 = 1/(4T_1^2)$), we have

$$\Pr[X_{T_1} \ge \varepsilon/4] \le \exp\left(-\frac{\varepsilon^2/16}{1/(2T_1) + \varepsilon/(12T_1)}\right) \le \exp\left(-\frac{\varepsilon^2 T_1}{33}\right) \le \frac{1}{2T}.$$

Note that

$$\frac{1}{T_1} \sum_{t \in [T_1]} |b_t - v_t| = X_{T_1} + \frac{\varepsilon}{4}.$$

So

$$\Pr\left[\frac{1}{T_1} \sum_{t \in [T_1]} |b_t - v_t| \ge \frac{\varepsilon}{2}\right] = \Pr[X_{T_1} \ge \varepsilon/4] \le \frac{1}{2T}.$$

Finally, by the definition of the $W_1$ distance, with probability at least $1 - 1/(2T)$,

$$W_1(\bar{\mathcal{D}}, \hat{\mathcal{D}}) \le \frac{1}{T_1} \sum_{t \in [T_1]} |b_t - v_t| \le \frac{\varepsilon}{2}.$$

This is because $W_1(\bar{\mathcal{D}}, \hat{\mathcal{D}})$ is no larger than the cost of matching each $b_t$ to $v_t$. $\qquad \square$

*Proof of Lemma 5.* By Lemmas 4 and 3, and taking a union bound, with probability at least $1 - 1/T$,

$$W_1(\mathcal{D}, \bar{\mathcal{D}}) \le \frac{\varepsilon}{2} \quad \text{and} \quad W_1(\bar{\mathcal{D}}, \hat{\mathcal{D}}) \le \frac{\varepsilon}{2}.$$

Since $W_1$ is a metric, whenever the above is true

$$W_1(\mathcal{D}, \hat{\mathcal{D}}) \le W_1(\mathcal{D}, \bar{\mathcal{D}}) + W_1(\bar{\mathcal{D}}, \hat{\mathcal{D}}) \le \frac{\varepsilon}{2} + \frac{\varepsilon}{2} = \varepsilon. \qquad \square$$

*Proof of Lemma 6.* First observe that when the buyer uses the above bidding strategy, their expected value from the item received is

$$\mathbb{E}[v \cdot \mathbb{I}[v \ge \theta]] = \mathbb{E}[v \mid v \ge \theta] \cdot \Pr[v \ge \theta].$$

Moreover, by the choice of $\theta$, they have ROI slackness

$$\mathbb{E}[(v - \tau \cdot q) \cdot \mathbb{I}[v \ge \theta]] = \mathbb{E}[(v - \tau \cdot q) \mid v \ge \theta] \cdot \Pr[v \ge \theta] = 0.$$

So the above bidding strategy is feasible.

Now consider any feasible bidding strategy $b' : [0, 1] \to [0, 1]$. Moreover, without loss of generality assume $b'(v) \in \{0, 1\}$ for any $v \in [0, 1]$, i.e., the buyer buys the item if $b'(v) = 1$, and leaves otherwise. If $\Pr[b'(v) = 1] = 0$, then clearly the expected value from the item received under $b'$ is 0, which does not exceed that under $b$. Now suppose $\Pr[b'(v) = 1] > 0$. Because the buyer is ROI constrained,

$$\mathbb{E}[v \mid b'(v) = 1] \ge \tau \cdot q.$$

But then by the choice of $\theta$, it must be the case that

$$\Pr[b'(v) = 1] \le \Pr[v \ge \theta].$$

Again by the choice of $\theta$, for any $S \subseteq [0, 1]$ where $\Pr[v \in S] \le \Pr[v \ge \theta]$,

$$\mathbb{E}[v \cdot \mathbb{I}[v \in S]] \le \sup_{T \subseteq [0,1] : \Pr[v \in T] \le \Pr[v \ge \theta]} \mathbb{E}[v \cdot \mathbb{E}[v \in T]] = \mathbb{E}[v \cdot \mathbb{I}[v \ge \theta]].$$

This applies to $S = \{v \mid b'(v) \ge \theta\}$ too, so the expected value from the item received under $b'$ satisfies

$$\mathbb{E}[v \cdot \mathbb{I}[b'(v) \ge \theta]] \le \mathbb{E}[v \cdot \mathbb{I}[v \ge \theta]].$$

In other words, $b$ is at least as good as $b'$. $\qquad \square$

*Proof of Lemma 7.* Recall that

$$\mathrm{OPT}_t(\mathcal{D}) = \mathop{\mathbb{E}}_{v \sim \mathcal{D}}[\max\{v/\tau_t - c_t, 0\}] = \mathop{\mathbb{E}}_{v \sim \mathcal{D}}[(v/\tau_t - c_t) \cdot \mathbb{I}[v/\tau_t \geq c_t]].$$

And similarly,

$$\mathrm{OPT}_t(\hat{\mathcal{D}}) = \mathop{\mathbb{E}}_{v \sim \hat{\mathcal{D}}}[\max\{v/\tau_t - c_t, 0\}] = \mathop{\mathbb{E}}_{v \sim \hat{\mathcal{D}}}[(v/\tau_t - c_t) \cdot \mathbb{I}[v/\tau_t \geq c_t]].$$

So we only need to show that

$$\mathop{\mathbb{E}}_{v \sim \mathcal{D}}[(v/\tau_t - c_t) \cdot \mathbb{I}[v/\tau_t \geq c_t]] - \mathop{\mathbb{E}}_{v \sim \hat{\mathcal{D}}}[(v/\tau_t - c_t) \cdot \mathbb{I}[v/\tau_t \geq c_t]] \leq \varepsilon.$$

Suppose otherwise, i.e., there is some $\varepsilon' > \varepsilon$ such that

$$\mathop{\mathbb{E}}_{v \sim \mathcal{D}}[(v/\tau_t - c_t) \cdot \mathbb{I}[v/\tau_t \geq c_t]] - \mathop{\mathbb{E}}_{v \sim \hat{\mathcal{D}}}[(v/\tau_t - c_t) \cdot \mathbb{I}[v/\tau_t \geq c_t]] = \varepsilon'.$$

Let $\mathcal{D}|_{\geq c_t}$ be such that for each $v \in [0, 1]$,

$$\mathop{\Pr}_{v' \sim \mathcal{D}|_{\geq c_t}}[v \geq v'] = \begin{cases} \Pr_{v' \sim \mathcal{D}}[v \geq v'], & \text{if } v/\tau_t \geq c_t \\ 0, & \text{otherwise.} \end{cases}$$

And construct $\hat{\mathcal{D}}|_{\geq c_t}$ from $\hat{\mathcal{D}}$ similarly. By construction we have

$$\mathop{\mathbb{E}}_{v \sim \mathcal{D}|_{\geq c_t}}[v/\tau_t - c_t] - \mathop{\mathbb{E}}_{v \sim \hat{\mathcal{D}}|_{\geq c_t}}[v/\tau_t - c_t] = \mathop{\mathbb{E}}_{v \sim \mathcal{D}|_{\geq c_t}}[v/\tau_t] - \mathop{\mathbb{E}}_{v \sim \hat{\mathcal{D}}|_{\geq c_t}}[v/\tau_t] = \varepsilon'.$$

Let $\mathcal{D}_j \in \Gamma(\mathcal{D}|_{\geq c_t}, \hat{\mathcal{D}}|_{\geq c_t})$ be such that

$$\mathop{\mathbb{E}}_{(v, \hat{v}) \sim \mathcal{D}_j}[|v - \hat{v}|] < \varepsilon'.$$

Note that $\mathcal{D}_j$ exists because by the definition of the $W_1$ distance,

$$W_1(\mathcal{D}|_{\geq c_t}, \hat{\mathcal{D}}|_{\geq c_t}) \leq W_1(\mathcal{D}, \hat{\mathcal{D}}) < \varepsilon'.$$

But then we have

$$\varepsilon' > \mathop{\mathbb{E}}_{(v, \hat{v}) \sim \mathcal{D}_j}[|v - \hat{v}|] \geq \left| \mathop{\mathbb{E}}_{(v, \hat{v}) \sim \mathcal{D}_j}[v - \hat{v}] \right| = \tau_t \cdot \left| \mathop{\mathbb{E}}_{v \sim \mathcal{D}|_{\geq c_t}}[v/\tau_t] - \mathop{\mathbb{E}}_{v \sim \hat{\mathcal{D}}|_{\geq c_t}}[v/\tau_t] \right| = \tau_t \cdot \varepsilon' \geq \varepsilon',$$

a contradiction. This concludes the proof. $\qquad\square$

*Proof of Lemma 8.* Let

$$\hat{q}_t = q_t + \frac{\varepsilon}{\Pr_{v \sim \hat{\mathcal{D}}}[v/\tau_t \geq c_t]} = \mathop{\mathbb{E}}_{v \sim \hat{\mathcal{D}}}[v/\tau_t \mid v/\tau_t \geq c_t].$$

Observe that

$$\tau_t \cdot c_t = \inf\{\theta \in [0, 1] \mid \mathop{\mathbb{E}}_{v \sim \hat{\mathcal{D}}}[v \mid v \geq \theta] \geq \tau \cdot \hat{q}_t\},$$

So by Lemma 6, the revenue from posting $\hat{q}_t$ is

$$
\begin{aligned}
(\hat{q}_t - c_t) \cdot \mathop{\Pr}_{v \sim \hat{\mathcal{D}}}[v \geq \tau_t \cdot c_t] &= \mathop{\mathbb{E}}_{v \sim \hat{\mathcal{D}}}[v/\tau_t - c_t \mid v \geq \tau_t \cdot c_t] \cdot \mathop{\Pr}_{v \sim \hat{\mathcal{D}}}[v \geq \tau_t \cdot c_t] &\text{(choice of } \hat{q}) \\
&= \mathop{\mathbb{E}}_{v \sim \hat{\mathcal{D}}}[\max\{v/\tau_t - c_t, 0\}] \\
&= \mathrm{OPT}_t(\hat{\mathcal{D}}). &\text{(definition of } \mathrm{OPT}_t(\hat{\mathcal{D}}))
\end{aligned}
$$

In other words, $\mathrm{OPT}_t(\hat{\mathcal{D}})$ is achieved by posting price $\hat{q}_t$ to the item when the value distribution is $\hat{\mathcal{D}}$.

Observe that $\mathrm{Rev}_t(\mathcal{D}, \hat{\mathcal{D}}) \geq 0$. So when $\mathrm{OPT}_t(\hat{\mathcal{D}}) < \varepsilon$, the lemma is trivial. Below we assume $\mathrm{OPT}_t(\hat{\mathcal{D}}) \geq \varepsilon$, which is equivalent to

$$(\hat{q}_t - c_t) \cdot \mathop{\Pr}_{v \sim \hat{\mathcal{D}}}[v \geq \tau_t \cdot c_t] \geq \varepsilon \iff q_t - c_t \geq 0.$$

For any $\varepsilon' > \varepsilon$, there exists $\mathcal{D}_j \in \Gamma(\mathcal{D}, \hat{\mathcal{D}})$ such that

$$\mathop{\mathbb{E}}_{(v,\hat{v}) \sim \mathcal{D}_j} [|v - \hat{v}|] = \varepsilon'.$$

$\mathcal{D}_j$ exists because $W_1(\mathcal{D}, \hat{\mathcal{D}}) \leq \varepsilon < \varepsilon'$. Then we have

$$
\begin{aligned}
\mathop{\mathbb{E}}_{(v,\hat{v}) \sim \mathcal{D}_j} [(v/\tau_t - q_t) \cdot \mathbb{I}[\hat{v} \geq \tau_t \cdot c_t]] &\geq \mathop{\mathbb{E}}_{(v,\hat{v}) \sim \mathcal{D}_j} [(\hat{v}/\tau_t - |v - \hat{v}|/\tau_t - q_t) \cdot \mathbb{I}[\hat{v} \geq \tau_t \cdot c_t]] \\
&= \text{OPT}_t(\hat{\mathcal{D}}) - \mathop{\mathbb{E}}_{(v,\hat{v}) \sim \mathcal{D}_j} [|v - \hat{v}| \cdot \mathbb{I}[\hat{v} \geq \tau_t \cdot c_t]]/\tau_t \\
&\qquad\qquad\qquad\qquad\qquad \text{(optimality of } \hat{q}_t \text{ for } \hat{\mathcal{D}}) \\
&\geq \text{OPT}_t(\hat{\mathcal{D}}) - \varepsilon'/\tau_t \qquad\qquad \text{(definition of } W_1(\mathcal{D}, \hat{\mathcal{D}})) \\
&\geq \text{OPT}_t(\hat{\mathcal{D}}) - \varepsilon'. \qquad\qquad\qquad\qquad (\tau_t \geq 1)
\end{aligned}
$$

Letting $\varepsilon' \to \varepsilon$, we have

$$\mathop{\mathbb{E}}_{(v,\hat{v}) \sim \mathcal{D}_j} [(v/\tau_t - q_t) \cdot \mathbb{I}[\hat{v} \geq \tau_t \cdot c_t]] \geq \text{OPT}_t(\hat{\mathcal{D}}) - \varepsilon.$$

This means there exists $S \subseteq [0, 1]$ satisfying

$$\mathop{\Pr}_{v \sim \mathcal{D}} [v \in S] = \mathop{\Pr}_{v \sim \hat{\mathcal{D}}} [v \geq \tau_t \cdot c_t],$$

such that

$$\mathop{\mathbb{E}}_{v \sim \mathcal{D}} [v/\tau_t - q_t \mid v \in S] \cdot \mathop{\Pr}_{v \sim \hat{\mathcal{D}}} [v \geq \tau_t \cdot c_t] \geq \text{OPT}_t(\hat{\mathcal{D}}) - \varepsilon = (\hat{q}_t - c_t) \cdot \mathop{\Pr}_{v \sim \hat{\mathcal{D}}} [v \geq \tau_t \cdot c_t] - \varepsilon.$$

Since $\Pr_{v \sim \hat{\mathcal{D}}}[v \geq \tau_t \cdot c_t] > 0$, the above is equivalent to

$$\mathop{\mathbb{E}}_{v \sim \mathcal{D}} [v/\tau_t - q_t \mid v \in S] \geq \hat{q}_t - \frac{\varepsilon}{\Pr_{v \sim \hat{\mathcal{D}}}[v \geq \tau_t \cdot c_t]} - c_t = q_t - c_t \geq 0.$$

In other words, $S$ satisfies

$$\mathop{\mathbb{E}}_{v \sim \mathcal{D}} [v \mid v \in S] \geq \tau_t \cdot q_t \quad \text{and} \quad \mathop{\Pr}_{v \sim \mathcal{D}} [v \in S] = \mathop{\Pr}_{v \sim \hat{\mathcal{D}}} [v \geq \tau_t \cdot c_t].$$

Now by the choice of $\theta_t$, we must have

$$\mathop{\Pr}_{v \sim \mathcal{D}} [v \geq \theta_t] \geq \mathop{\Pr}_{v \sim \mathcal{D}} [v \in S] = \mathop{\Pr}_{v \sim \hat{\mathcal{D}}} [v \geq \tau_t \cdot c_t].$$

So the revenue of the exploitation mechanism can be bounded in the following way:

$$\text{Rev}_t(\mathcal{D}, \hat{\mathcal{D}}) \geq (q_t - c_t) \cdot \mathop{\Pr}_{v \sim \hat{\mathcal{D}}} [v \geq \tau_t \cdot c_t] = (\hat{q}_t - c_t) \cdot \mathop{\Pr}_{v \sim \hat{\mathcal{D}}} [v \geq \tau_t \cdot c_t] - \varepsilon = \text{OPT}_t(\hat{\mathcal{D}}) - \varepsilon.$$

This concludes the proof. $\qquad\square$

*Proof of Lemma 9.* By Lemma 8,

$$\text{Rev}_t(\mathcal{D}, \hat{\mathcal{D}}) \geq \text{OPT}_t(\hat{\mathcal{D}}) - \varepsilon.$$

And by Lemma 7,

$$\text{OPT}_t(\hat{\mathcal{D}}) \geq \text{OPT}_t(\mathcal{D}) - \varepsilon.$$

Putting the above inequalities together, the lemma follows immediately. $\qquad\square$

*Proof of Theorem 1.* By Lemma 5, with probability $1 - 1/T$, we have $W_1(\mathcal{D}, \hat{\mathcal{D}}) \leq \varepsilon$. Below we condition on this happening. For each $t \in [T]$, let

$$
R_t := \begin{cases}
(b_t/\tau_t - c_t) \cdot x_t, & \text{if } t \in [T_1] \\
0, & \text{if } t \in \{T_1 + 1, \ldots, T_1 + T_2\} \\
\max\{(q_t - c_t), 0\} \cdot \mathbb{I}[v_t \geq \theta_t], & \text{otherwise.}
\end{cases}
$$

be the realized revenue in stage $t$, where $\mathbb{E}_{v_t}[R_t] = \mathrm{Rev}_t(\mathcal{D}, \hat{\mathcal{D}})$ for each $t \in \{T_1 + T_2 + 1, \ldots, T\}$. By Lemma 9, for each $t \in \{T_1 + T_2 + 1, \ldots, T\}$, $\mathrm{Rev}_t(\mathcal{D}, \hat{\mathcal{D}}) \geq \mathrm{OPT}_t(\mathcal{D}) - 2\varepsilon$, so

$$
\begin{aligned}
\mathbb{E}_{\{v_t\}_t} \left[ \sum_{t \in [T]} R_t \right] &\geq \mathbb{E}_{\{v_t\}_t} \left[ \sum_{T_1+T_2+1 \leq t \leq T} R_t \right] - (T_1 + T_2) && (c_t \leq 1 \text{ for all } t) \\
&\geq \sum_{T_1+T_2+1 \leq t \leq T} \mathrm{OPT}_t(\mathcal{D}) - (T_1 + T_2) - T \cdot 2\varepsilon && (\text{Lemma 9}) \\
&\geq \sum_{t \in [T]} \mathrm{OPT}_t(\mathcal{D}) - 2(T_1 + T_2) - T \cdot 2\varepsilon && (\mathrm{OPT}_t(\mathcal{D}) \leq 1 \text{ for all } t \in [T]) \\
&= \mathrm{OPT}(\mathcal{D}) - 2^{19} T^{2/3} \log T.
\end{aligned}
$$

Observe that $\{R_t\}_t$ are independent conditioning on all randomness from the exploration stage (because $R_t$ depends only on $v_t$), and that $0 \leq R_t \leq 1$ for each $t \in \{T_1 + T_2 + 1, \ldots, T\}$. So by Hoeffding's inequality,

$$
\begin{aligned}
&\Pr_{\{v_t\}_t} \left[ \sum_{T_1+T_2+1 \leq t \leq T} R_t \leq (\mathrm{OPT}(\mathcal{D}) - 2^{19} T^{2/3} \log T) - T^{1/2} \log T \right] \\
&\leq \Pr_{\{v_t\}_t} \left[ \sum_{T_1+T_2+1 \leq t \leq T} R_t \leq \sum_{T_1+T_2+1 \leq t \leq T} \mathrm{Rev}_t(\mathcal{D}, \hat{\mathcal{D}}) - T^{1/2} \log T \right] \\
&\leq \exp\left(-2 \log T\right) = \frac{1}{T^2}.
\end{aligned}
$$

Finally, taking a union bound, the probability that the exploration stage succeeds and that the realized revenue $\sum_t R_t$ concentrates around the mean $\sum_t \mathrm{Rev}_t$ is at least $1 - 2/T$, and whenever this happens, the revenue collected by our mechanism is at least $\mathrm{OPT}(\mathcal{D}) - O(T^{2/3} \log T)$. This finishes the proof. $\qquad\square$