# OpenReview forum: "Prior-independent Dynamic Auctions for a Value-maximizing Buyer"
_NeurIPS.cc/2021/Conference — NeurIPS 2021 Poster_

### Official Review · Reviewer_giSA · 2021-07-13

**Rating:** 6
**Confidence:** 3

**Summary:**

This paper studies how a seller should design its auctions to maximize her revenue when facing an impatient buyer optimizing his value under a ROI constraint. It mostly consists in plugging together two previous works, Balseiro 2021 and Amin 2013 to extend Balseiro 2021 to the setting where the value distribution is unknown a priori and needs to be learnt. Following Amin 2013, the proposed algorithm has two phases (an explore then commit schema) and the first phase is designed to make him reveal his value distribution, which allows to implement an optimal mechanism in the second phase. Provided that the buyer is impatient enough, the first phase length can be tuned to incur an overall sub-linear regret.

**Ethical Concerns:**

Nothing comes to my mind.

**Limitations And Societal Impact:**

see main review

**Main Review:**

**Originality**
* The algorithm, the result and the main line of proof are a well-executed but not surprising combination of results from Balseiro 2021 and Amin 2013
  * The mechanism used in exploration phase is an instantiation of a mechanism satisfying Balseiro 2021 Lemma 3.2
  * The one used in exploitation phase is a small variation of the optimal mechanism from Balseiro 2021 Mechanism 4.3
  * The design and line of proof of the phased algorithm follows Amin 2013.
* The technical contributions consist in
  * the way to measure the drift between the true empirical distribution and the observed bids with the Wasserstein distance, which allows to use literature on Wassertein concentration and martingale tail-inequalities,
  * modifying the optimal mechanism in phase 2 to account for the (small) misreporting coming from the first phase.

**Quality and clarity**
The algorithm is well presented and explained. The paper is clear, and easy to read. I checked most of the proofs.

**Significance**
* I have some concern about the ROI constraint as presented in the paper. In Balseiro 2021, they use a ROI constraint in expectation as they are studying a single-shot auction. This paper studies a repeated setting and thus it would make more sense for the ROI constraint to be cumulative over time. Indeed, by only enforcing a ROI constraint in expectation at each time step, cumulatively, the ROI constraint could be violated by as much as $\sqrt(T)$ I think. This paper makes the argument that each $t$ corresponds to a period of time (a day), but it would mean the buyer provides a single bid each day, which is not always very realistic.
* I think it should be clear from the title and the abstract that the paper only looks at "impatient" buyers, as for instance, there is no lower bound to characterize what happens is $\gamma$ is close to 1 while $\tau$ is known.

**Minor details**
* line 136 (end of line), I think you meant "buyer" rather than "seller"
* In the appendix, you could use another notation for derivatives of univariate functions, as using "prime" for both derivatives and indexing several variables like $v$ and $v'$ is kind of confusing. Maybe, you could use the "dot" notation for derivative, e.g. $\dot{x}(v')$ rather than $x'(v')$ which is kind of confusing at first.

**Time Spent Reviewing:**

6

---

> ### Author Response · Authors · 2021-08-10
> **Author Response to Reviewer giSA**
>
> Thank you for your thoughtful and encouraging comments.  Regarding cumulative ROI constraints: this is definitely an interesting model too, and we will discuss it as a future direction.  As for in-expectation ROI constraints, one can simulate "many bids in the same day" by making many copies of a stage with the same target ROI ratio, where the buyer can submit different bids.  In that case, a "day" corresponds to a number of consecutive stages which all share the same target ROI ratio.  We will discuss this in the paper.
>
> Regarding impatient buyers: we are happy to change the abstract and the title to reflect the fact that the buyer is impatient (if there is a consensus among the reviewers -- we don't want to slip unrefereed major changes into the paper).
>
> "seller" should be "buyer": thanks for catching this.  We will fix it.
>
> Dot for derivative: thanks for the suggestion.  We will make changes accordingly.

---

### Official Review · Reviewer_JHak · 2021-07-14

**Rating:** 6
**Confidence:** 3

**Summary:**

The paper studies prior-independent dynamic auctions with a single value-maximizing buyer.
In this setting, the buyer's objective is to maximize her obtained value (rather than utility) under ROI constraint (i.e., the value is at least $\tao$ times higher than the price paid).
The authors show a dynamic auction mechanism that without having no prior information on the buyer’s value distribution (except from being bound in [0,1]), can extract a near-optimal revenue, i.e., sublinear regret for time horizon T.
The work is under the assumption of a patient seller and a buyer has a time discount factor $\lambda$.

The algorithm is divided into three parts:

1. Exploration phase that incentivizes the buyer to report approximately truthfully. To my understanding, this is the main new technique in the work.

2. A buffer phase.

3. Exploitation phase that extracts the revenue using a robust version of the single-stage revenue-optimal mechanism for this problem.



**Limitations And Societal Impact:**

Since the focus of this work is primarily theoretical, any societal impact depends on the application.

**Main Review:**

Learning to maximize revenue without knowing the prior is well motivated and was studied in the standard context of a utility-maximizing buyer.
Extending those results to the framework of a value-maximizing buyer (under ROI or any other constraint) is a natural research goal.

I am not completely sure about the results’ significance, since the algorithm and proof techniques seem rather standard. The framework of a value-maximizing buyer under ROI constraint was introduced in previous work, so the conceptual contribution also seems limited.

Comments and questions:
What is the known lower bound for regret in this model?

The work assumes the values are drawn from a distribution bounded in [0,1].
Do you have results for distributions bound in [0,d] for $d$ unknown to the seller?
Do you have results for unbounded distributions?

Line 136: the second seller in the line should be a buyer?

In the definition of the buyer optimization problem (line 123 and in general), the buyer’s objective depends on the future (with a discount) while the constraint only depends on the current step. Shouldn’t the ROI constraint consider future decisions as well? Especially since the constraint only cares about the expectation and not the realization at this specific time.



**Time Spent Reviewing:**

6

---

> ### Author Response · Authors · 2021-08-10
> **Author Response to Reviewer JHak**
>
> Thank you for your thoughtful and encouraging comments.  Regarding lower bounds: there is no known lower bound in the model studied in this paper (though the T^{2/3} regret is likely optimal among two-phase exploration-exploitation mechanisms for the following reason: the "error" of the exploration phase normally decreases at rate 1/\sqrt{t}, where t is the length of the phase, and this error is the regret incurred in every exploitation stage.  So the optimal length of the exploration phase is about t = T^{2/3}, giving overall regret T^{2/3}.).  Techniques for proving lower bounds in similar models may not easily apply here due to different benchmarks and incentive models.  We will discuss this as a future direction.
>
> Values in [0, d] / unbounded values: the regret (being additive) generally has to depend on d.  For values in [0, d], our mechanism can be easily adapted to give regret d * T^{2/3}.  This linear dependence on d is indeed necessary: suppose we have a family of mechanisms M_d for values in [0, d] for any d, whose regret does not depend on d (or depends sublinearly on d -- here we assume no dependence for simplicity).  Then we can adapt it to obtain a mechanism with arbitrarily small regret for values in [0, 1], which is impossible.  To see why this is the case, consider the following mechanism for values in [0, 1] constructed from M_d: when receiving a bid b (wlog b is in [0, 1]), let b' = d * b, and feed b' to M_d.  Let x' and p' be the allocation and payment output by M_d.  The actual allocation and payment we choose will then be x = x' and p = p' / d.  The regret of this mechanism is precisely the regret of M_d scaled down by a factor of d.  Taking d to infinity, we obtain a mechanism for values in [0, 1] with arbitrarily small regret.
>
> "seller" should be "buyer": thanks for catching this.  We will fix it.
>
> Regarding "cumulative" ROI constraint: conceptually, a stage in our model may correspond to the time period of a "campaign", which has a single target ROI ratio.  The model with a "cumulative" ROI constraint certainly makes sense too, and we will discuss it as a future direction.

---

### Official Review · Reviewer_5rjq · 2021-07-14

**Rating:** 7
**Confidence:** 4

**Summary:**

This paper studies the problem of a seller who interacts repeatedly with a buyer.  In each round an item is up for sale, and the buyer draws a value from an unknown (to the seller) distribution.  The buyer is assumed to be less patient than the seller, and the seller's goal is to maximize revenue.  Prior work had established low-regret sales strategies for the seller when the buyer is a utility maximizer.  This paper considers a value-maximizing bidder, who wants to maximize value obtained subject to an interim ROI constraint.

The authors construct a dynamic mechanism with regret O(T^{2/3}), versus the best revenue attainable if the value distribution were known.  At a high level the approach is similar to the utility-maximizing case: run an explore phase that obtains low revenue but learns about the buyer's value distribution, then deploy a mechanism aimed at extracting near-optimal revenue from what was learned.  The key here is to incentivize the buyer not to misreport in the explore phase, which is done by using a strictly truthful per-round auction (and leveraging the assumption that the buyer is less patient).  For a value-maximizing buyer, what is needed for strict truthfulness is different.  So a main technical contribution of the work is to propose a new single-round auction format that is strictly single-stage incentive compatible.



**Limitations And Societal Impact:**

I would recommend more discussion motivating the choice to have tau publicly revealed in each round.  The paper currently just cites previous works that make the same choice, but further discussion on this point would be very helpful.

**Main Review:**

The authors do a nice job of explaining the barriers to extending prior work on prior-free dynamic mechanisms to the setting of value maximization.  The new construction is clever and natural.  There are quite a few subtleties to the analysis and the paper covers them quite thoroughly.  Overall I found the presentation quite clear.

One downside to the result is that the mechanism requires knowledge of the buyer's target ROI ratio tau in each round, which by assumption can change over time.  So it is assumed that while an agent's value is part of her private type, the ROI constraint is not.  I found this a little hard to motivate, since both value and ROI-constraint both form the underlying preferences of the buyer.  Previous work on value-maximizing bidders have also assumed public ROI constraints, so this assumption has precedence, but even so I would have liked to see some discussion about how necessary it is.

Conceptually, the approach taken by this paper shares a lot in common with prior work.  There are certainly details to work out, and I appreciated the care taken to show how to extend the construction to this more general setting.  But nevertheless the marginal contribution does not feel especially exciting, and even the new single-round mechanism (Proposition 1) is not very far from known methods for utility-maximizing agents, given that the ROI parameter tau is known and can be used explicitly.

Overall, this feels like a solid advance in the development of dynamic prior-independent auctions for an interesting set of bidders.  The marginal contribution is a bit slight, but I feel the work is above the bar for NeurIPS and can be accepted if there is room.

===

Post author response: I've read the response, and I am still favorable toward acceptance.

**Time Spent Reviewing:**

2

---

> ### Author Response · Authors · 2021-08-10
> **Author Response to Reviewer 5rjq**
>
> Thank you for your thoughtful and encouraging comments.  Regarding public ROI ratio: indeed, as the reviewer points out, it is not clear how to handle private value + private ROI ratio even in static environments, which means it's not even clear what the benchmark should be (first best may not be achievable any more).  We will discuss this in the paper.

---

### Official Review · Reviewer_FsWG · 2021-07-16

**Rating:** 7
**Confidence:** 4

**Summary:**

This paper considers the dynamic mechanism design problem to maximize total expected revenue against a single bidder, who aims to maximize time-discounted cumulative expected value, while subject to an interim ROI constraint each period. The work proposes a dynamic mechanism consisting of an exploration phase during which the seller runs an incentive-compatible mechanism to estimate the bidder’s value distribution; and also an exploitation phase that approximates the single-stage optimal mechanism against an ROI bidder w.r.t. the distributional estimates. The paper shows that the design limits the magnitude of the bidder’s misreporting, and proves that the proposed dynamic mechanism achieves a $T$-period regret in the order of $T^{2/3}$ against the optimal revenue of a clairvoyant seller who knows the bidder’s value realizations.

**Limitations And Societal Impact:**

The paper did not adequately address limitations in terms of the model choice and proposed mechanisms. For detailed suggestions see main review above.

**Main Review:**

I enjoyed reading the paper, as the idea of “exploring” through incentive compatible mechanisms to learn the bidder’s value distribution, and then “exploiting” via incentive-compatible revenue-optimal mechanisms for ROI constrained bidders using such distributional estimates, is interesting. Although previous works established the technique of leveraging bidder time discounted utilities to bound the deviation from truthful bidding, to the best of my knowledge adopting this methodology in a dynamic mechanism design context is novel. In addition, the paper is well-written and the key results are presented in a succinct, yet informative, fashion.

My only concern for this submission is that it does not comment on the optimality of the $T^{2/3}$ regret achieved. For comprehensiveness, it would be nice to include some statement on, or at least some intuition into, the regret lower bound. Along this line regarding bidder discount factors, Amine et al.,2013 demonstrated that in the repeated posted price setting, if the discount factor is 1 (i.e. no discount), sublinear regret is impossible. I am curious if a similar argument is directly applicable to the dynamic mechanism design setting of interest. It would be nice if the authors can include relevant discussions in the paper.

In my opinion, there are two limitations in terms of modelling choices:

1.The paper studies per-period ROI constraints, which I think greatly simplifies the problem since it decouples the mechanisms from period to period, and only requires the seller approximate the single-period optimal mechanism for ROI constrained bidders (which is an established result in previous work). Moreover, I think in practice ROI is typically measured using the total value divided by total expenditure across an entire ad campaign. Hence, from both a theoretical and practical perspective, perhaps a more interesting model would be to impose a single ROI constraint for the entire time horizon.

2.The paper assumes the bidder knows her value distribution, which may be an unrealistic
assumption from a practical viewpoint. On the other hand, if the bidder is also learning her distribution, then the seller may possibly exploit the bidder's learning algorithm and achieve higher revenue compared to the clairvoyant benchmark.

Nevertheless, I do not think these limitations of the model undermine the contributions/key insights of this paper, and perhaps the authors can comment on relevant aspects in a “future directions” section (if space permits).


**Time Spent Reviewing:**

5

---

> ### Author Response · Authors · 2021-08-10
> **Author Response to Reviewer FsWG**
>
> Thank you for your thoughtful and encouraging comments.  Regarding the optimality of T^{2/3}: the bound is likely optimal (up to polylog factors) among two-phase exploration-exploitation methods in the following sense: the "error" of the exploration phase normally decreases at rate 1/\sqrt{t}, where t is the length of the phase, and this error is the regret incurred in every exploitation stage.  So the optimal length of the exploration phase is about t = T^{2/3}, giving overall regret T^{2/3}.  In general we don't know if one can do better by, for example, combining exploration and exploitation and gradually decreasing the target error rate -- which would require fundamentally different single-stage mechanisms as building blocks (it's not clear how to achieve both exploration and exploitation even for the far more well-studied case of utility-maximizing buyers).  We also remark that the proof techniques by Amin et al. may not easily generalize to our model due to different benchmarks and incentive models.  We will discuss this in the paper.
>
> Regarding modelling choices: these are indeed interesting aspects, and we will discuss them as future directions.  We remark that people don't know how to handle some of these even in static environments (e.g., unknown value + unknown ROI ratio).

---

> > ### Comment · Reviewer_FsWG · 2021-08-31
> > **Re:**
> >
> > Thanks for your response and clarifications. I have read other reviews and responses, and my opinions about the paper have not changed.

---

### Decision · Program_Chairs · 2021-09-27

**Decision:**

Accept (Poster)

**Comment:**

The reviews are generally positive, although not enthusiastic. There are some pros and cons. My reading of the main issues in the reviews and the discussion is as follows:

[+] The explore-then-exploit mechanism is quite technical. This is somewhat unusual: many explore-then-exploit algorithms or mechanisms in the literature are often "the easy part" of the paper, followed either by a more complicated approach based on "adaptive exploration" (and $\sqrt{T}$ regret), or a lower bound which rules it out.

[+] As the authors suggest, $T^{2/3}$ regret appears optimal among all explore-then-exploit mechanisms. Such lower bound seems within reach, via a standard lower-bounding technique for bandits (*), and would be a natural complement to the main result. I encourage the authors to try to prove it.

[-]  Long-term incentives are reduced to short-term incentives in a simple, brute-force way: a $O(\log T)$-sized "padding" is inserted between exploration and exploitation, to make the entire exploitation worth very little to the buyer.

[-] This brute-force approach is feasible because the seller is assumed to be infinitely patient. This approach would probably fail if the seller had a  larger but comparable time-discount factor. Which is arguably closer to reality: e.g., advertiser's time scale is week or even a month, whereas the platform's time scale is a year.

[+/-] The novelty lies in the design of the single-shot mechanism for exploration (which needs to be incentive compatible in a very strong sense: large deviation should cause a large loss in buyer's utility), and perhaps also the single-shot mechanism  for exploitation (which needs to be robust to uncertainty in the value distribution $D$). However, one reviewer suggests that both single-shot mechanisms are similar to those in Balseiro et al (2021).

[-] Some of the other simplifications are not that well-justified: that ROI constraint is per-round rather than aggregate, and that target ROI is public. Probably not a big flaw, though.

(*) It may be possible to "reduce" directly to the respective lower bound for bandits. Else, it would be a slightly more involved problem-specific argument. A similar argument (for explore-then-exploit mechanisms in a technically different auction design setting) can be found in (Moshe Babaioff, Yogeshwer Sharma, Aleksandrs Slivkins: Characterizing Truthful Multi-armed Bandit Mechanisms, EC 2009, SICOMP 2014) and (Nikhil R. Devanur, Sham M. Kakade: The price of truthfulness for pay-per-click auctions. EC 2009).